# Ceftriaxone Mediated Synthesized Gold Nanoparticles: A Nano-Therapeutic Tool to Target Bacterial Resistance

**DOI:** 10.3390/pharmaceutics13111896

**Published:** 2021-11-08

**Authors:** Farhan Alshammari, Bushra Alshammari, Afrasim Moin, Abdulwahab Alamri, Turki Al Hagbani, Ahmed Alobaida, Abu Baker, Salman Khan, Syed Mohd Danish Rizvi

**Affiliations:** 1Department of Pharmaceutics, College of Pharmacy, University of Ha’il, Ha’il 81442, Saudi Arabia; frh.alshammari@uoh.edu.sa (F.A.); afrasimmoin@yahoo.co.in (A.M.); t.alhagbani@uoh.edu.sa (T.A.H.); a.alobaida@uoh.edu.sa (A.A.); 2Department of Medical Surgical Nursing, College of Nursing, University of Ha’il, Ha’il 81442, Saudi Arabia; bu.alshammari@uoh.edu.sa; 3Department of Pharmacology and Toxicology, College of Pharmacy, University of Ha’il, Ha’il 81442, Saudi Arabia; a.alamry@uoh.edu.sa; 4Nanomedicine and Nanotechnology Lab, Department of Biosciences, Integral University, Lucknow 226026, India; karimabubaker@gmail.com (A.B.); salmank@iul.ac.in (S.K.)

**Keywords:** antibiotics, antibacterial resistance, ceftriaxone, gold nanoparticles, MIC_50_

## Abstract

Ceftriaxone has been a part of therapeutic regime for combating some of the most aggressive bacterial infections in the last few decades. However, increasing bacterial resistance towards ceftriaxone and other third generation cephalosporin antibiotics has raised serious clinical concerns especially due to their misuse in the COVID-19 era. Advancement in nanotechnology has converted nano-therapeutic vision into a plausible reality with better targeting and reduced drug consumption. Thus, in the present study, gold nanoparticles (GNPs) were synthesized by using ceftriaxone antibiotic that acts as a reducing as well as capping agent. Ceftriaxone-loaded GNPs (CGNPs) were initially characterized by UV-visible spectroscopy, DLS, Zeta potential, Electron microscopy and FT-IR. However, a TEM micrograph showed a uniform size of 21 ± 1 nm for the synthesized CGNPs. Further, both (CGNPs) and pure ceftriaxone were examined for their efficacy against *Escherichia coli*, *Staphylococcus aureus, Salmonella abony* and *Klebsiella pneumoniae*. CGNPs showed MIC_50_ as 1.39, 1.6, 1.1 and 0.9 µg/mL against *E. coli*, *S. aureus*, *S. abony* and *K. pneumoniae*, respectively. Interestingly, CGNPs showed two times better efficacy when compared with pure ceftriaxone against the tested bacterial strains. Restoring the potential of unresponsive or less efficient ceftriaxone via gold nanoformulations is the most alluring concept of the whole study. Moreover, applicability of the findings from bench to bedside needs further validation.

## 1. Introduction

According to Centers for Disease Control and Prevention (CDC) threat report on antibiotic resistance (2019) [1], every year around 2.8 million cases of resistant bacterial infection occur in the United States alone with 35 K mortality. However, COVID-19 co-infection with antibiotic-resistant bacterial pathogens has raised a serious clinical issue now-adays. The situation has worsened due to the increasing trend of self-medication of antibiotics in the COVID-19 era [2]. One such antibiotic is ceftriaxone, and scientists have grave concerns over the cautious use of antibiotics in COVID-19 management [3]. In fact, ceftriaxone is often prescribed to treat a wide range of bacterial infections, such as meningitis, bone infections, joints, middle ear, intra-abdominal, skin, and pelvic inflammatory diseases [4]. On the other hand, ceftriaxone resistance has also increased many fold in the recent past [5,6,7]. Thus, alternative novel approaches to tackle the current scenario are urgently needed, and nanotechnology appears to deliver a plausible solution to these resistant issues.

Application of nanoparticles in different fields of medicine has gained worldwide acceptance because of their unique physical and bio-chemical features, and controlled drug release ability [8]. In the past few years, several inorganic nanoparticles with antibacterial potential have been developed, such as gold nanoparticles, silver nanoparticles, zinc oxide nanoparticles, and titanium dioxide nanoparticles [9,10,11,12,13,14,15]. These inorganic nanoparticles can inhibit bacterial growth by various mechanisms such as hindering replication and transcription process, DNA damage via direct interaction, increasing reactive oxygen species, destroying the cell wall, etc. [13]. Importantly, they have shown effectiveness against resistant bacterial strains [14,15].

Among the different inorganic nanoparticles, gold nanoparticles (GNPs) are of major interest in diverse research fields such as therapeutics, antimicrobials, catalysis, and biomolecular detection [16,17,18]. A two-fold increase in antibiotic activity was observed when ampicillin was conjugated with chitosan-capped GNPs, compared to free ampicillin [9]. GNPs capped with Human Serum Albumin (HSA) have been used for the successful delivery of antibiotics of the amino-glycosidic group, such as sulfates of streptomycin, neomycin, gentamicin, and kanamycin [19]. Similarly, sericin-capped silver and GNPs have shown marked activity against both Gram-negative and Gram-positive bacteria [20]. In addition, silver and GNPs have shown the ability to overcome ampicillin and cefaclor resistance [21], although, GNPs are considered more biocompatible and safe than silver nanoparticles [22,23].

All these findings incited us to explore solution(s) for expanding ceftriaxone resistance via applying GNPs as a nano-carrier. The thought behind the current study is to increase ceftriaxone strength by loading them onto GNPs. The study involved synthesis of GNPs by employing ceftriaxone as a reducing and capping agent, and to boost ceftriaxone antibacterial potential in a coordinated manner against *Escherichia coli*, *Staphylococcus aureus*, *Salmonella abony* and *Klebsiella pneumoniae*.

## 2. Materials and Methods

### 2.1. Materials

All the chemicals, microbiological media and reagents, such as, Mueller–Hinton agar, ceftriaxone sodium and tetra chloroauric [III] acid (HAHuCl_4_) were procured from Sigma–Aldrich (St. Louis, MO, USA).

### 2.2. Ceftriaxone-Mediated Synthesis of GNPs (CGNPs)

The reaction for the synthesis of CGNPs was performed at temperatures of 30 °C, 40 °C, 50 °C and 60 °C by adding ceftriaxone at concentrations of 50, 100, 150, 200, and 250 µg/mL to a 1 mL reaction mixture and incubating for 48 h. However, the reaction mixture consisted of 1 mM HAuCl_4_ in 50 mM phosphate buffer at pH 7.4.

An autonomous reaction was performed for the control without ceftriaxone. At ten distinct time points, the mixture was removed and analyzed by UV-visible spectroscopy. The CGNPs were collected by centrifugation at 30,000× *g* for 30 min after the completion of the reaction. The CGNPs were then washed twice by Milli Q water followed by a final wash with 50% *v*/*v* ethanol to remove unbounded materials. For further analysis, the resultant CGNPs were used. For comparative analysis, the Khan et al. [24] method was applied to synthesize GNPs by bromelain (where bromelain acts as a reducing as well as stabilizing agent) to keep them as control naked GNPs (without ceftriaxone).

### 2.3. Characterization of CGNPs

#### 2.3.1. UV/Vis Spectroscopy

The GNPs (control) and CGNPs were characterized via UV-vis spectrophotometry using a Shimadzu dual-beam spectrophotometer (UV-1601 PC, Shimadzu, Tokyo, Japan) at a resolution of 1 nm.

#### 2.3.2. Dynamic Light Scattering (DLS)

The mean particle size of GNPs and CGNPs was measured with a DLS particle size analyzer (Zetasizer Nano-ZS, Malvern Instrument Ltd., Malvern, UK). The sample was taken in a DTS0112-low volume disposable sizing cuvette of 1.5 mL. The sample was sonicated for 1 min and filtered through syringe membrane filters with pores less than 0.45 µm before measurement. The mean particle size was determined by calculating the average of the measurements of a single sample in triplicate. Zeta potential was also measured to observe the nature of charge present on the surface of each sample by using Zetasizer Nano-ZS, Malvern Instrument Ltd., Malvern, UK. For zeta potential, DTS1070 disposable cuvette was used.

#### 2.3.3. Scanning Electron Microscopy (SEM)

A drop from each sample, GNPs and CGNPs solutions, was deposited onto a conductive silicon substrate and dried on a hotplate at 60 °C for 20 min. The morphology of deposited GNPs and CGNPs on Si substrates were then imaged using FEI quanta 250 SEM (FEI Company, Hillsboro, OR, USA) with an accelerating voltage of 30 KV and a spot size of 3 nm.

#### 2.3.4. Transmission Electron Microscopy (TEM)

TEM was performed using a Tecnai™ G2 Spirit Bio-TWIN equipped with a CCD camera (GatanDigital, Hillsboro, OR, USA). CGNPs sample was prepared using a carbon-coated TEM copper grid.

#### 2.3.5. Fourier Transform Infrared Spectroscopy (FTIR)

FTIR (Shimadzu FTIR-8201 PC, PerkinElmer Inc., Waltham, MA, USA) was used to analyze the binding conformation and changes in secondary structures on the ceftriaxone present on the surface of CGNPs. The instrument was operated in diffuse reflectance mode at a resolution of 4 cm^−1^ to obtain good signal-to-noise ratios, 256 scans of CGNP films were obtained at a range of 400–4000 cm^−1^.

### 2.4. Ceftriaxone Loading Efficiency on GNPs

#### 2.4.1. Calculation by UV-Vis Spectrophotometry

The ceftriaxone loading efficacy onto GNPs was evaluated by using the methodology of Gomes et al. [25] as applied in Shaikh et al. [26]. Once the CGNPs were synthesized (without washing), the samples were centrifuged at 30,000× *g* for 30 min. Ceftriaxone in the supernatant was quantified by using UV-Visible spectrophotometer (λmax 241) after scanning [27,28]. However, the 5–70 μg/mL concentration range was used to plot calibration curve of ceftriaxone. For evaluating ceftriaxone loading efficacy, free ceftriaxone present in the supernatant was subtracted from the initial amount added for the CGNPs synthesis. The following equation was used to evaluate the % of loading efficacy:$$\mathrm{Percentage}\;\mathrm{of}\;\mathrm{loading}\;\mathrm{efficacy}=\frac{\left[\mathrm{Amt}.\;\mathrm{of}\;\mathrm{ceftriaxone}\;\mathrm{used}\;\left(\mathrm{Total}\right)-\mathrm{Free}\;\mathrm{ceftriaxone}\;\mathrm{in}\;\mathrm{supernatant}\right]}{\mathrm{Amt}.\;\mathrm{of}\;\mathrm{ceftriaxone}\;\mathrm{used}\;\left(\mathrm{Total}\right)}\times 100$$

#### 2.4.2. Calculation by High Performance Liquid Chromatography (HPLC)

The loading efficacy of ceftriaxone onto GNPs was also estimated by using the modified methodology of Pal et al. [29]. Shimadzu HPLC model fitted with UV/VIS detector (SPD-20A), AT pump (LC-20) and rheodyne injector with a 20-µL loop were used. Samples were analyzed on a reverse phase C-18 (Luna −5 µm, 250 × 4.6 mm inner diameter) column at 25 °C by applying a mobile phase (0.01 M KH2PO4:ACN buffer in 85:15 ratio) with 1 mL/min flow rate and UV-detection at 241 nm. Spinchrom software was used to record and evaluate the data. Before analyzing, a 0.22 µm filter was used to filter the mobile phase. Each sample was run in triplicate, and a calibration curve was plotted by using 5–70 μg/mL concentration of ceftriaxone. The amount of unbound ceftriaxone was calculated by using the calibration curve, and the amount of capped ceftriaxone onto GNPs was calculated by subtracting the unbound ceftriaxone from the total amount of ceftriaxone added. The exact amount of capped ceftriaxone was calculated using the following equation:$$\mathrm{Percentage}\;\mathrm{of}\;\mathrm{drug}\;\mathrm{capping}=\frac{\left[\mathrm{Amt}.\;\mathrm{of}\;\mathrm{ceftriaxone}\;\mathrm{capped}\right]}{\mathrm{Amt}.\;\mathrm{of}\;\mathrm{ceftriaxone}\;\mathrm{used}\;\left(\mathrm{Total}\right)}\times 100$$

### 2.5. Antibacterial Activity Evaluation

#### 2.5.1. Bacteria and Growth Conditions

*Escherichia coli* (ATCC 25922), *Klebsiella pneumoniae* (ATCC 13883), *Salmonella abony* (NCIM 2257) and *Staphylococcus aureus* (ATCC 25923) were obtained from National Chemical Laboratory, India. Luria–Bertani (LB) broth was used to prepare fresh inoculum for each bacterial strain and incubated at 37 °C for 18 h. Prior to antibacterial activity, LB broth was used to adjust the turbidity of culture to 0.5 McFarland standard i.e., 1.5 × 10^8^ CFU/mL.

#### 2.5.2. Qualitative Assessment of Antibacterial Activity

Before performing the antibacterial assay, solutions were prepared by dispersing the synthesized CGNPs, GNPs (control) and ceftriaxone in PBS (phosphate saline buffer at pH 7.4). The agar well diffusion method was applied to assess the potency of synthesized CGNPs [30]. Fresh bacterial culture for each strain was spread on Mueller–Hinton agar and 6 mm wells were cut on 1 mg/mL) and GNPs (control) were dispensed in the wells. All the experiments were performed in triplicate, and the agar plates were placed in an incubator at a temperature of 37 °C overnight. The diameter of the zone of inhibition was measured.

#### 2.5.3. Determination of the MIC

The synthesized CGNPs and ceftriaxone were tested against bacterial strains to determine their minimum inhibitory concentrations (MICs) by employing the broth microdilution method of Eloff [31]. To achieve the concentrations ranging from 0.025–32 µg/mL, aliquots of CGNPs and ceftriaxone were serially diluted in 96-well microtiter plates containing LB broth medium. The tested strains were cultured overnight in LB broth, and their turbidity was adjusted to 0.5 McFarland standard (1.5 × 10^8^ CFU/mL), following these plates. A total of 50 µL of CGNPs (200 µg/mL ceftriaxone), ceftriaxone (which, 10 µL of the standard suspensions was placed in the aliquots. MICs are the lowest concentrations of synthesized CGNPs that completely inhibit bacterial growth after being incubated at 37 °C for 20 h.

## 3. Results and Discussion

### 3.1. CGNPs Synthesis

Several biomolecules and chemicals have been utilized as capping and reducing agents in the synthesis of multi-purpose inorganic nanoparticles [32]. Generally, conjugation of antibiotic/drug is performed on pre-formed GNPs by using different strategies. GNPs are synthesized either by chemicals (such as sodium borohydrate and trisodium citrate) or by herbal extracts and natural enzymes before conjugating antibiotics onto them [26,33,34,35]. In both the cases, residual contamination might create a doubt on the actual antibacterial results.

Typically, gold salt reduction followed by nucleation and nuclei growth leads to the synthesis of GNPs, and synthesized GNPs need a capping agent to be stabilized [36,37,38]. The highlight of the present study is that ceftriaxone acted as both reducing and capping/stabilizing agents for the synthesis of (ceftriaxone loaded gold nanoparticles) CGNPs (Figure 1). It is a fact that by changing the concentration of reducing agent (especially when it acts as a reducing as well as capping agent) and experimental conditions, the size of GNPs can be controlled [36,37,38]. Here, the different concentrations of ceftriaxone along with different temperature conditions were applied to synthesize CGNPs. Finally, the 250 µg/mL ceftriaxone concentration was selected to reduce HAuCl_4_ to GNPs to obtain the desired size at a temperature of 40 °C and pH of 7.4. Khan et al. [24] and Khan et al. [39] have also applied the same strategy to synthesize GNPs of various sizes using bromelain and trypsin as reducing and capping agents. Similarly, the properties such as size, shape, mono-dispersity and stability of CGNPs in the present study basically relied on ceftriaxone concentration and temperature used for the reaction (data not shown for brevity). The synthesized GNPs and CGNPs showed visible characteristic color changes from yellow to ruby red (Appendix A).

### 3.2. Characterization of CGNPs

#### 3.2.1. Spectrophotometric

Typical ‘Surface Plasma Resonance’ band patterns for synthesized gold nanoformulations were characterized using UV-Visible spectroscopy. GNPs (control/without ceftriaxone showed absorption λmax at 520 nm, while CGNPs showed maximum absorption at 536 nm (Figure 2). The red shift of absorption from 520 to 536 nm can be correlated with the changes in size that might have occurred due to attachment of ceftriaxone to the CGNPs [40,41]. In a 2017 study, Shaikh et al. [26] also observed the same red shift after the attachment of cefotaxime antibiotic to the GNPs. However, they conjugated the antibiotic on preformed GNPs instead of synthesizing them by the one-pot synthesis method that has been developed during the present study. During CGNP spectrophotometric analysis, an additional peak at 241 nm was also detected that corresponds to ceftriaxone attached to CGNPs [27]. Similarly, other studies have also shown two peaks when antibiotics (secnidazole-320 nm and cefotaxime-298 nm) were conjugated to GNPs along with characteristic peaks of 525 [42] and 542 nm [26]. In fact, it has been observed that the capping agent has a major influence on the electrocatalytic activity of GNPs [43].

#### 3.2.2. Dynamic Light Scattering (DLS) and Electron Microscopy

Z-average size by DLS for GNPs and CGNPs was estimated as 51.59 and 95.07 nm, respectively (Figure 3). The size by DLS is based on the details of inner inorganic core of nanoparticles along with the solvent layer that has adhered to the nanoparticles once they are disseminated in the liquid medium. Thus, relying only on DLS is not enough to know the actual size of inorganic core. Zeta potential of GNPs and CGNPs was found to be −16.6 and −25.7 mV, respectively, which is an indicator of good stability of both the nanoparticles [44]. Usually, larger zeta potential either −ve or +ve implicates much more stable dispersion, that means nanoparticles will not get aggregated due to repulsion between each other [45,46]. However, emulsion and colloid stability are not always predicted by zeta-potential, as only repulsive electrostatic forces are measured, and the forces of attraction such as Van der Waals forces are not considered [47]. Thus, the stability was also checked by keeping the colloidal CGNPs at room temperature for months and no aggregation was found even after 5 months.

Scanning Electron Microscopy (SEM) results showed that both GNPs and CGNPs were spherical in shape and monodispersed (Figure 4). Ceftriaxone attachment/capping has not caused any changes in the shape of GNPs. In accordance, several other reports have also suggested the similar spherical pattern of GNPs after antibiotic conjugation [26,42].

Transmission Electron Microscopy (TEM) has been performed for GNPs and CGNPs to estimate the size of the inorganic core. Using the TEM analysis by Gatan Digital Micrograph (Figure 5), the size of the GNPs and CGNPs were confirmed to be 10.2 ± 1 and 21 ± 1 nm, respectively. The optical properties of GNPs were accredited to the 5 d (valence) and 6 sp (conduction) electrons. Well-defined monodispersed nanoparticles of equal size were revealed by the TEM micrograph. Estimating size by TEM and DLS covers two different aspects. DLS provide size distribution and polydispersity index results based on the quantification of several million particles present in the colloidal form, while TEM results are considered more biased in terms of selective imaging, where only a few hundred particles could be quantified at one time. Thus, correlating both the approaches has become an important strategy worldwide.

### 3.3. FTIR Spectra of CGNPs and Ceftriaxone

Confirmation of the interactions between the surface of gold nanoparticles and ceftriaxone was done by FTIR spectroscopy (Figure 6a,b). The FTIR spectrum of ceftriaxone shows chief absorption bands at 3426 and 3265 cm^−1^. The emergence of the aforementioned absorption bands indicates the stretching vibrations in the N–H and O–H groups, respectively. The absorption band at 2935 cm^−1^ indicates stretching band vibrations of C–H groups, range between 1741 and 1650 cm^−1^ is designated for the stretching vibrations of the carbonyl group (C=O), and the absorption band corresponding to 1538 cm^−1^ indicates torsional vibrations of the aromatic ring.

Absorption bands corresponding to 1383 and 1034 cm^−1^ indicate the stretching vibration values of the C–N and C–O bonds. However, the interaction of ceftriaxone with the surface of gold nanoparticles causes the merging of the absorption bands and reduces the absorption intensities in C=O, N-H, and O-H groups. The absorption intensities of C=O, N-H, and O-H are 1798–1637, 3422, and 3265 cm^−1^, respectively.

### 3.4. Calculation of Loading Efficiency

Prior to antibacterial assessment, loading efficiency of ceftriaxone on GNPs was calculated by UV-Visible spectrophotometric and HPLC method. Here, 199.8 μg of ceftriaxone (by UV-Visible spectrophotometry) and 199.5 μg of ceftriaxone (by HPLC) was found to be loaded to the GNPs, out of 250 μg of the ceftriaxone initially used for the synthesis. Thus, the loading efficiency percentage was estimated as 79.92% and 79.80%, respectively, for the methods used. Furthermore, the retention time for pure ceftriaxone and capped ceftriaxone is estimated as 3.512 min (Figure 7a) and 3.59 min (Figure 7b), respectively. The observable slight change in retention time was might be due to variation of pH in the mobile phase to the medium of the drug. The retention time for CGNPs is 2.61 min (Figure 7b). Similarly, in a 2015 study, secnidazole was estimated by HPLC, and found to have 70% loading efficacy onto GNPs [42]. In another study, cefotaxime loading efficacy on GNPs was found as 77.59% when estimated by UV-Visible spectrophotometry [26]. It is a fact that higher loading efficiency correlates inversely with unwanted loss of antibiotic/drug and shows better therapeutic application [25]. Therefore, the methodology applied in the present study was effective in loading a good amount of ceftriaxone onto gold nanoparticles. However, 200 μg ceftriaxone was considered as the final loaded amount on GNPs as an approximation for further antibacterial assay to avoid difficulties in calculations.

### 3.5. Antibacterial Activity of CGNPs 

The antibacterial activities of GNPs (Control), CGNPs and ceftriaxone were evaluated by testing them against three gGram-negative strains, i.e., *Escherichia coli*, *Salmonella abony* and *Klebsiella pneumoniae* and one Gram-positive *Staphylococcus aureus*. These tested strains were chosen to represent different bacterial types of machinery nurturing several potent virulent factors other than their observable pathogenicity and their prevalence in day-to-day life. The promising detection revealed that CGNPs and ceftriaxone could inhibit the growth of bacteria after diffusion into the agar (Appendix A). Also, it was observed that both CGNPs and pure ceftriaxone had similar zones of inhibition. However, the total concentration of ceftriaxone in 50 µL CGNPs was equivalent to only 10 µg/well, whereas, the concentration of pure ceftriaxone was 50 µg/well. Thus, our primary findings confirmed that effectiveness of CGNPs was higher than pure ceftriaxone.

The MIC_50_ of CGNPs and pure ceftriaxone against all the tested bacterial strains was recorded (Figure 8). The MIC_50_ values for GNPs and pure ceftriaxone were 1.39 and 3.1 µg/mL against *E. coli* (Figure 8a)*,* 1.60 and 2.9 µg/mL against *S. aureus* (Figure 8b), 1.1 and 2.07 µg/mL against *S. abony* (Figure 8c), 0.9 and 2.4 µg/mL against *K. pneumoniae* (Figure 8d), respectively. 

Based on the antibacterial results, it can be suggested that ceftriaxone attachment to gold nanoparticles has enhanced its potency twice than the pure ceftriaxone. GNPs without ceftriaxone were used as a control and they did not show any activity against any tested strain. Similar results were observed when Shaikh et al. [26] and Brown et al. [48] tested naked GNPs while studying the cefotaxime- and ampicillin-conjugated GNPs against resistant bacterial strains, respectively. Thus, it can be inferred that the activity was due to ceftriaxone, and GNPs just aided in augmenting the potency. Due to biocompatibility, non-cytotoxicity and exceptional physiochemical properties, gold nanoformulations have always been the first choice among inorganic nanoparticles for drug delivery [22,23]. Importantly, it was observed that the reactive portion of antibiotic (ciprofloxacin) was surface exposed when it was attached to GNPs and activity is retained [49]. Our results were in harmony with the findings of Shaikh et al. [26] and Brown et al. [48], where cefotaxime and ampicillin also retained their potency after conjugation to GNPs.

The hypothesis on the mechanistic aspect of CGNP antibacterial action is based on the earlier reports of Rai et al. [21] and Shaikh et al. [26]. Firstly, the effective delivered ceftriaxone concentration was increased due to its attachment to GNPs. It might be due to the typical properties of GNPs, i.e., high surface-to-volume ratio, high concentration of (ceftriaxone) molecules loaded onto it due to large surface area, increased permeability towards the biological membrane and higher uptake by the bacterial cell [50]. Secondly, GNPs might have increased the porosity of the targeted bacterial strains and ceftriaxone molecules have gained easy access to the bacterial cell for their action. In fact, increased delivered concentration of antibiotic could saturate the resistant enzymes such as beta-lactamses, and plausibly inhibit the growth of the beta-lactamase-containing resistant bacterial strains as well [26]. However, when we discuss human cellular uptake of the nanoparticles (within the nanometers size range), pinocytosis is considered as a major uptake mechanism [51]. In fact, pinocytosis is a continual process occurring in all the cells that could be subdivided as clathrin-mediated endocytosis, micropinocytosis, clathrin- and caveolae-independent endocytosis, and caveolae-mediated endocytosis [52,53]. It has been observed that if the size is below 100 nm, the pinocytosis uptake mechanism is preferred, whereas, if the size is large (250 nm), phagocytosis occurs [54,55]. In our study, the size of both GNPs and CGNPs (as observed by TEM) are appropriate for pinocytosis. Although, further studies are needed to pinpoint the exact pinocytosis mechanism followed by CGNPs for the cellar uptake.

The most persistent global public health issue after COVID-19 is antimicrobial resistance due to the resultant restriction in therapeutic options against infections, and misuse/self-medication of antibiotics in the COVID-19 era [2]. Recently, novel strategies have been designed to enhance the properties (distribution, penetration, specificity, and pharmacokinetics) of antimicrobial drugs. The formulation of antimicrobial nanoparticles or antimicrobial-conjugated nanoparticles is one such strategy. Impressive increases in drug specificity and enhanced pharmacokinetics were observed when GNPs are utilized for antimicrobial delivery. In this study, a similar approach has been used to enhance the potency of ceftriaxone. Ceftriaxone resistance was globally accepted before the arrival of COVID-19, but it is speculated to increase with time as suggested by several reports. Thus, the solutions are warranted urgently. In our study, it has been found GNPs can markedly enhance the potency of ceftriaxone. Moreover, its fate in the human body and toxicity aspects still needed to be deciphered. Currently, our team is working on exploring the exact mechanism of action, toxicity and lethal dose of CGNPs using in-vivo and in-vitro experimental designs. On the basis of preliminary findings on toxicity (data not shown), we found no toxicity on normal cell lines. Our team hopes that we can come up with fresh nanoformulations to tackle bacterial resistance issues in the near future.

## 4. Conclusions

The present study delivered an approach to synthesize gold nanoparticles by applying ceftriaxone as reducing as well as stabilizing agent. In addition, synthesized ceftriaxone-loaded gold nanoparticles (CGNPs) acted as an effective tool to deliver ceftriaxone to the tested bacterial strains and markedly enhanced the ceftriaxone potency. Comparative analysis of pure ceftriaxone and CGNPs revealed that ceftriaxone after loading onto GNPs could become two times more potent. This strategy has opened a path to synthesize and deliver different antibiotics through GNPs in a one-step process to resolve the issue of increasing resistance. However, in-vivo studies to evaluate the fate and toxicity of CGNPs are warranted before jumping into the conclusive statement on the applicability of synthesized nanoformulations. Moreover, the preliminary findings of the present study could be used as a base to develop applicable nanoformulations.

## Figures and Tables

**Figure 1 pharmaceutics-13-01896-f001:**
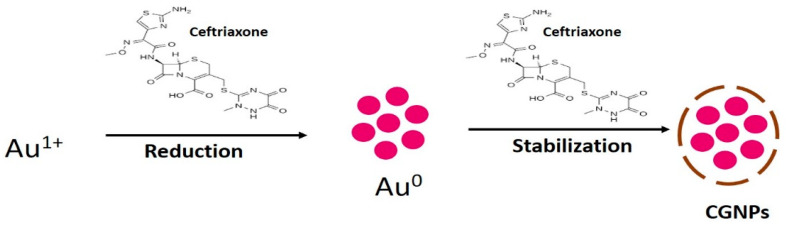
Scheme of ceftriaxone-mediated synthesis of gold nanoparticles. Here, CGNPs are the ceftriaxone-stabilized gold nanoparticles.

**Figure 2 pharmaceutics-13-01896-f002:**
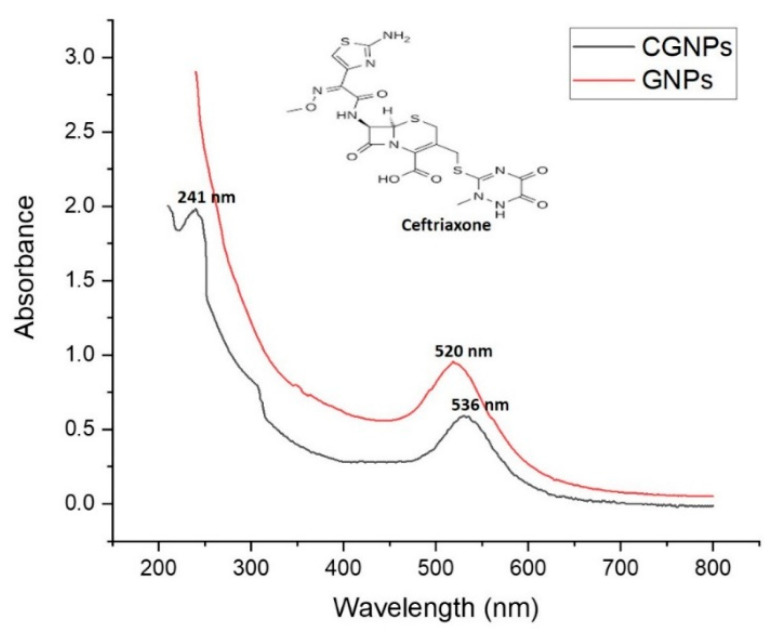
UV-Visible spectrophotometric characterization of GNPs and CGNPs.

**Figure 3 pharmaceutics-13-01896-f003:**
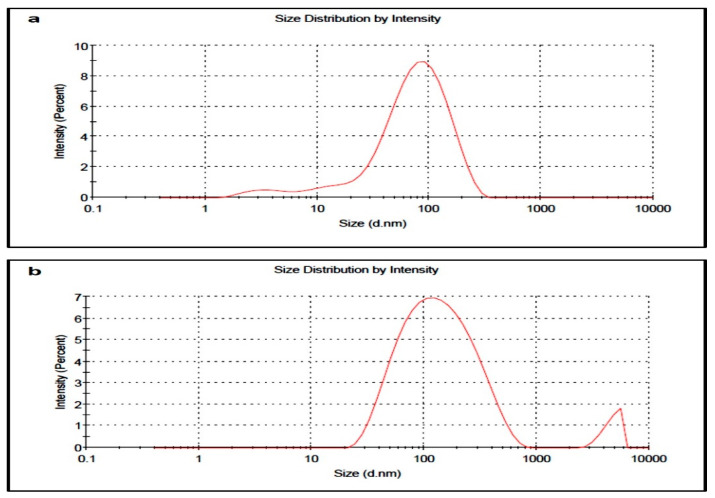
Z-average particle size of (**a**) GNPs and (**b**) CGNPs measured by DLS.

**Figure 4 pharmaceutics-13-01896-f004:**
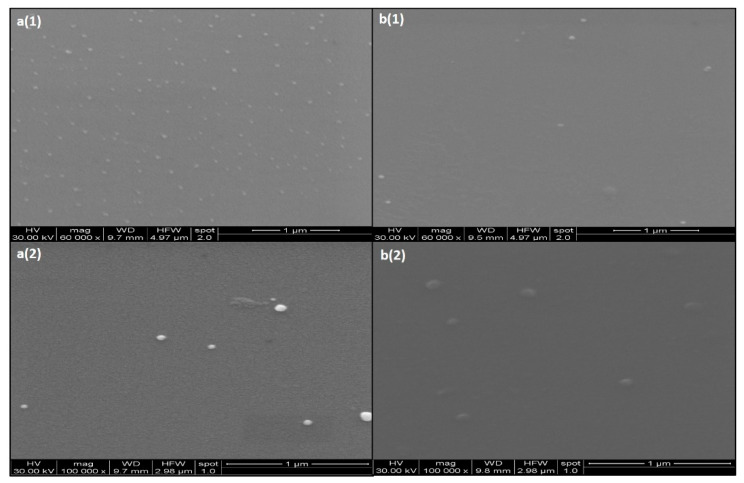
Scanning electron images of (**a1**,**a2**) GNPs at different magnifications and (**b1**,**b2**) CGNPs at different magnifications.

**Figure 5 pharmaceutics-13-01896-f005:**
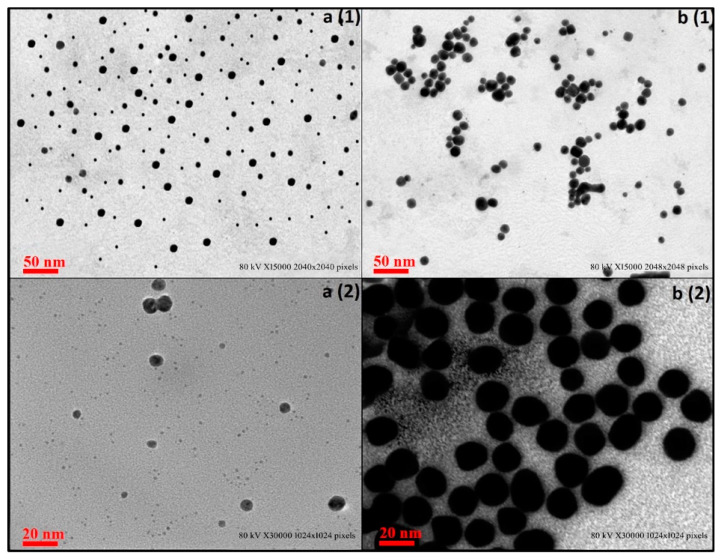
TEM Micrograph images of GNPs (**a1**,**a2**) and CGNPs (**b1**,**b2**).

**Figure 6 pharmaceutics-13-01896-f006:**
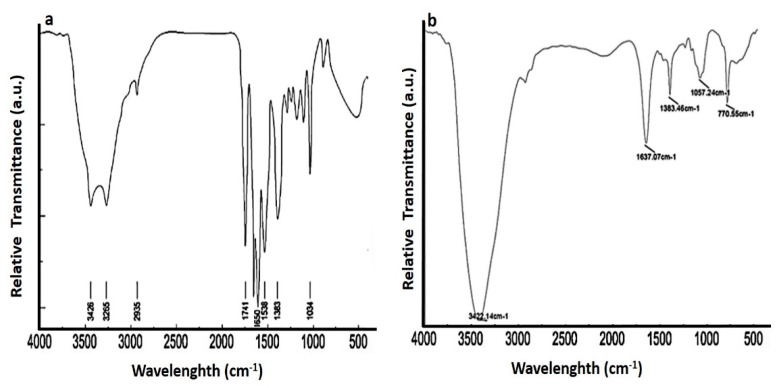
FTIR spectra of (**a**) pure ceftriaxone and (**b**) CGNPs.

**Figure 7 pharmaceutics-13-01896-f007:**
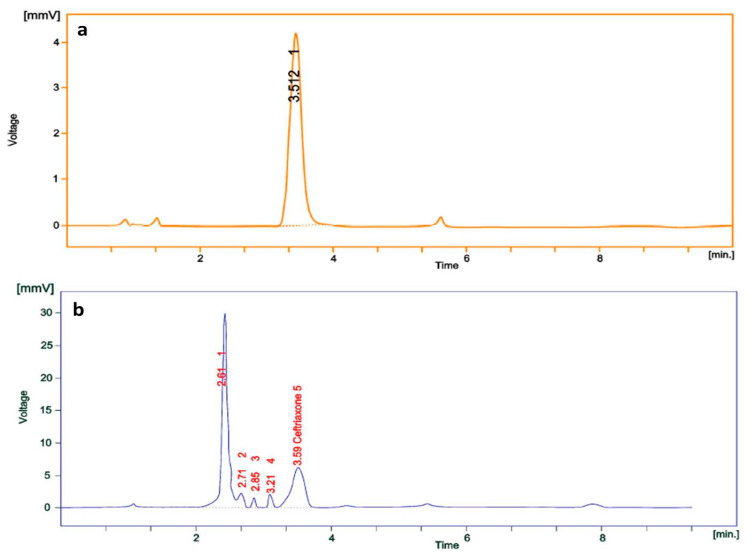
RP-HPLC chromatogram of (**a**) pure ceftriaxone and (**b**) CGNPs.

**Figure 8 pharmaceutics-13-01896-f008:**
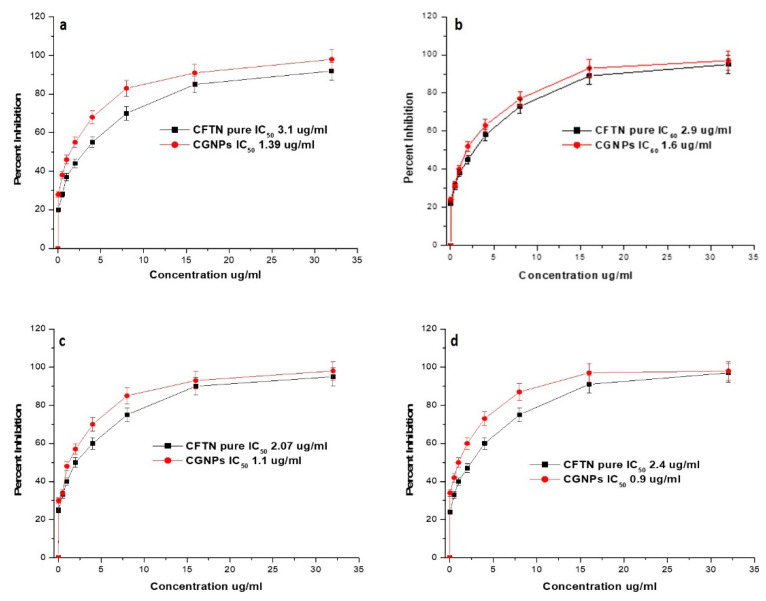
Determination of Minimum Inhibitory Concentration (MIC) of CGNPs and ceftriaxone (CFTN) against (**a**) *Escherichia coli*, (**b**) *Staphylococcus aureus*, (**c**) *Salmonella abony*, and (**d**) *Klebsiella pneumoniae*.

## Data Availability

Not applicable.

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
