# Peer review of "Ceftriaxone Mediated Synthesized Gold Nanoparticles: A Nano-Therapeutic Tool to Target Bacterial Resistance"

_pharmaceutics, 2021, doi:10.3390/pharmaceutics13111896_

Round 1
Reviewer 1 Report
The manuscript presents the synthesis and characterization of Ceftriaxone capped gold nanoparticles, and the in vitro analysis of its antibacterial activity. The topic is interesting but is not novel [Iraqi Journal of Science 2015, 56, 2425-2438; J. Microbiol. Biotechnol. 2018, 28(9), 1563-1572; Annals of R.S.C.B. 2021, 25, 6814 - 6821]. Anyway, the manuscript could be suitable for publishing in this journal after major revision, considering the following comments.
- Correct the word Cefriaxone in the title (i.e. Ceftriaxone).
- Introduction:
- CDC should be defined before to be used (i.e. Centers for Disease Control and Prevention).
- A reference (e.g. a review) should be included after the affirmation: “Application of nanoparticles in different fields of the medicine have gained worldwide acceptance because of their unique physical and bio-chemical features, and controlled drug release ability”. For instance: [Curr. Opin. Biotechnol. 2015, 35, 135-140].
-More recent references should be included after the sentence: “Among the different inorganic nanoparticles, gold nanoparticles (GNPs)… and biomolecular detection [15]”. For instance: [10.1007/s41061-019-0273-0] and [10.3390/nano10122340] both from 2020.
- Materials and Methods:
- The protocol for the synthesis of gold nanoparticles without Ceftriaxone (i.e. GNPs) should reported, specifying the reducing and stabilizing ligand.
- DLS should be used in the paragraph of the section 2.2.2.
-Similarly, SEM should be used in the paragraph of the section 2.2.3.
- Results and Discussion:
- Atibiotics should be corrected (i.e. antibiotics) in the first paragraph of the section 3.2.1 (page 5).
- An inset Figure should be included in Figure 2, showing the normalized absorbance of the resonance plasmon bands from both NPs (i.e. GNPs and CGNPs). The band displacement should be commented like in this reference providing AuNPs capped with different ligands [10.1039/C9TA05492H].
-The format of Figure 3 should be the same used in Figures 2, 6 and 7.
-Lack of an analysis of the long-term stability of the resulting GNPs and CGNPs. For instance, as reported previously in this paper with inorganic nanoparticles for biomedical applications using DLS and Zeta-potential in different aqueous media: [10.1039/C6NR07462F].
-Figure 4-caption: The magnifications are not necessary because the scale bar is in the SEM images.
-SEM-EDX mapping could be included to show the different capping ligand compositions.
-Figure 5 should include a TEM image of the GNPs without Ceftriaxone to compare with CGNPs.
-Figure 6 should be improved because it seems distorted. The numbers in axes X have different sizes. Also, the numbers in the axe Y should be removed and the axe Y titled as relative transmittance (a.u.).
-Bar errors should be showed in the experimental points included in Figure 7. In addition, the analysis of GNPs and its reducing agents should be added as control samples in this study. This information is very interesting for the expert readers in this research field.
-The different internalization mechanisms (such as, Phagocytosis, Endocytosis, etc.) could be discussed (i.e. Ceftriaxone versus CGNPs and versus bare GNPs).
-Figure S1 should be include GNPs without CFTN to show the antibacterial activity of these inorganic nanoparticles against these gram negative bacterial.
-A photo of vials containing both Au colloidal dispersions (GNPs and CGNPs) could be added in the Supporting Information.
Author Response
Reply to reviewer’s comments
First of all, we appreciate the time spend by the honorable reviewers from their busy schedule to improve the quality of our MS. All the suggestions given by the reviewers have been addressed and highlighted in Yellow.
Reviewer 1:
- Correct the word Cefriaxone in the title (i.e. Ceftriaxone).
Response: We apologize for our mistake; the typographical error has been corrected in the revised MS.
Introduction:
- CDC should be defined before to be used (i.e. Centers for Disease Control and Prevention).
Response: We have duly defined the full form of CDC in the revised MS.
- A reference (e.g. a review) should be included after the affirmation: “Application of nanoparticles in different fields of the medicine have gained worldwide acceptance because of their unique physical and bio-chemical features, and controlled drug release ability”. For instance: [Curr. Opin. Biotechnol. 2015, 35, 135-140].
Response: Núñez-Lozano R, Cano M, Pimentel B, de la Cueva-Méndez G. 'Smartening' anticancer therapeutic nanosystems using biomolecules. Curr Opin Biotechnol. 2015 Dec;35:135-40. doi: 10.1016/j.copbio.2015.07.005. Epub 2015 Aug 13. PMID: 26277646.
- More recent references should be included after the sentence: “Among the different inorganic nanoparticles, gold nanoparticles (GNPs)… and biomolecular detection [15]”. For instance: [10.1007/s41061-019-0273-0] and [10.3390/nano10122340] both from 2020.
Response: As per the suggestion of honorable reviewer, the following updated references has been duly added in the revised MS.
- Alba-Molina, D., Giner-Casares, J.J. & Cano, M. Bioconjugated Plasmonic Nanoparticles for Enhanced Skin Penetration. Top Curr Chem (Z)378, 8 (2020).
- Fuster MG, Montalbán MG, Carissimi G, Lima B, Feresin GE, Cano M, Giner-Casares JJ, López-Cascales JJ, Enriz RD, Víllora G. Antibacterial Effect of Chitosan–Gold Nanoparticles and Computational Modeling of the Interaction between Chitosan and a Lipid Bilayer Model. Nanomaterials. 2020 Dec;10(12):2340.
Materials and Methods:
- The protocol for the synthesis of gold nanoparticles without Ceftriaxone (i.e. GNPs) should reported, specifying the reducing and stabilizing ligand.
Response: In the protocol, bromelain act as reducing as well as capping agent, which has been duly incorporated in the revised MS. It is noteworthy to mention that we have filed a patent as well for the same protocol.
M.S.Khan, S.M.Rizvi, Official Journal of the Patent Office (Indian Patent) 2014. Issue No.12/2014.
- DLS should be used in the paragraph of the section 2.2.2.
Response: We have duly incorporated DLS in the section 2.2.2.
- Similarly, SEM should be used in the paragraph of the section 2.2.3.
Response: We have duly incorporated SEM in the section 2.2.3.
Results and Discussion:
- Atibiotics should be corrected (i.e. antibiotics) in the first paragraph of the section 3.2.1 (page 5).
Response: Typographical error has been duly corrected in the revised MS.
- An inset Figure should be included in Figure 2, showing the normalized absorbance of the resonance plasmon bands from both NPs (i.e. GNPs and CGNPs). The band displacement should be commented like in this reference providing AuNPs capped with different ligands [10.1039/C9TA05492H].
Response: As per the suggestion of honorable reviewer, the following reference has been duly added to further justify our results in the revised MS.
- Alba-Molina D, Santiago AR, Giner-Casares JJ, Rodríguez-Castellón E, Martín-Romero MT, Camacho L, Luque R, Cano M. Tailoring the ORR and HER electrocatalytic performances of gold nanoparticles through metal–ligand interfaces. Journal of materials chemistry A. 2019 Sep 10;7(35):20425-34.
- The format of Figure 3 should be the same used in Figures 2, 6 and 7.
Response: As per the suggestion of the honorable reviewer, Figure 3 has been duly modified in revised MS.
- Lack of an analysis of the long-term stability of the resulting GNPs and CGNPs. For instance, as reported previously in this paper with inorganic nanoparticles for biomedical applications using DLS and Zeta-potential in different aqueous media: [10.1039/C6NR07462F].
Response: We have already stated that the nanoparticles are stable for months at room temperature and showed the zeta potential of the synthesized gold nanoparticles. However, to further justify the stability, the following reference has been duly added.
- Cano M, Núñez-Lozano R, Lumbreras R, González-Rodríguez V, Delgado-García A, Jiménez-Hoyuela JM, de la Cueva-Méndez G. Partial PEGylation of superparamagnetic iron oxide nanoparticles thinly coated with amine-silane as a source of ultrastable tunable nanosystems for biomedical applications. Nanoscale. 2017;9(2):812-22.
- Figure 4-caption: The magnifications are not necessary because the scale bar is in the SEM images.
Response: As per the suggestion of honorable reviewer, the figure 4 caption has been duly modified.
- SEM-EDX mapping could be included to show the different capping ligand compositions.
Response: We agree with the suggestion of honorable reviewer, we actually outsource for the SEM analysis, and did not go for SEM-EDX mapping this time. Adding it now is not feasible for us due to time constrain and current situation. However, we will definitely add it in our future work. Most humbly, we request honorable reviewer to consider only SEM images for the current MS.
- Figure 5 should include a TEM image of the GNPs without Ceftriaxone to compare with CGNPs.
Response: As per the suggestion of honorable reviewer, TEM images of GNPs without ceftriaxone are added in Figure 5 (in addition to two different scales of 20nm and 50nm).
- Figure 6 should be improved because it seems distorted. The numbers in axes X have different sizes. Also, the numbers in the axe Y should be removed and the axe Y titled as relative transmittance (a.u.).
Response: As per the suggestion of honorable reviewer, Figure 6 has been improved and added in the revised MS.
- Bar errors should be showed in the experimental points included in Figure 7. In addition, the analysis of GNPs and its reducing agents should be added as control samples in this study. This information is very interesting for the expert readers in this research field.
Response: As per the suggestion of the honorable reviewer, the error bars have been duly added in the figure 7. GNPs (without ceftriaxone) synthesized by bromelain were used as a control and did not showed any effect on the growth of bacteria. Shaikh et al. 2017 report similar findings, where bromelain synthesized GNPs do not show any antibacterial effect. The same has been highlighted in the revised MS as well.
- Shaikh, S.; Rizvi, S.M.D.; Shakil, S.; Hussain, T.; Alshammari, T.M.; Ahmad, W.; Tabrez, S.; Al-Qahtani, M.H.; Abuzenadah, A.M. Synthesis and Characterization of Cefotaxime Conjugated Gold Nanoparticles and Their Use to Target Drug-Resistant CTX-M-Producing Bacterial Pathogens. J Cell Biochem 2017, 118(9), 2802-2808.
- The different internalization mechanisms (such as, Phagocytosis, Endocytosis, etc.) could be discussed (i.e. Ceftriaxone versus CGNPs and versus bare GNPs).
Response: As per the suggestion of honorable reviewer, some statements based on earlier researches have been added to predict the internalization mechanisms with the following references.
- Zhao F, Zhao Y, Liu Y, Chang X, Chen C, Zhao Y (2011) Cellular uptake, intracellular trafficking, and cytotoxicity of nanomaterials. Small 7(10):1322–1337.
- Yu Y (2018) Resolving Endosome Rotation in Intracellular Trafficking. Biophys J 114(3, Supplement 1):630a
- Hillaireau H, Couvreur P (2009) Nanocarriers’ entry into the cell: relevance to drug delivery. Cell Mol Life Sci 66(17):2873–2896
- Panariti A, Miserocchi G, Rivolta I (2012) The effect of nanoparticle uptake on cellular behavior: disrupting or enabling functions? Nanotechnol Sci Appl 5:87
- Foroozandeh, P., Aziz, A.A. Insight into Cellular Uptake and Intracellular Trafficking of Nanoparticles. Nanoscale Res Lett 13, 339 (2018).
- Figure S1 should be include GNPs without CFTN to show the antibacterial activity of these inorganic nanoparticles against these gram negative bacterial.
Response: We appreciate the concern of the honorable reviewer, however, GNPs (without ceftriaxone) was the control in the present study and it showed no effect on the growth at the tested concentration. The same has been highlighted in the Figure S1 caption.
- A photo of vials containing both Au colloidal dispersions (GNPs and CGNPs) could be added in the Supporting Information.
Response: As per the suggestion of honorable reviewer, Figure S1 has been added with vial of GNPs and CGNPs with characteristic ruby red color. However, Figure S1 has been replaced as Figure S2.

Reviewer 2 Report
The work entitled “Cefriaxone mediated synthesized gold nanoparticles: a nano therapeutic tool to target bacterial resistance” by Alshammari et al., 2021 has been reviewed. The work described the utilization of Cefriaxone in synthesis of gold nanoparticles to target the bacterial resistance. The concept of using the antibiotic drug as capping and reducing agent for the synthesis of gold nanoparticles and improving it therapeutic efficiency. But the characterization and antibacterial assay results are not sufficient to consider publication in international journal like Pharmaceuticals.
Specific comments:
- Section 2.3. It is essential to quantify the concentration of Cefriaxone in gold nanoparticles by HPLC.
- NMR analysis is required to evidence the capping and conjugation of Cefriaxone in gold nanoparticles?
- SEM analysis results is poor, it need to redo with high magnification.
- In case SEM presented in different magnification but why TEM showed only one picture ?
- Why the author not presented EDS results, AED, XRD with size measuring calculation.
- FTIR analysis not clearly stated the formation of the NPS
Author Response
Reply to reviewer’s comments
First of all, we appreciate the time spend by the honorable reviewers from their busy schedule to improve the quality of our MS. All the suggestions given by the reviewers have been addressed and highlighted in Yellow.
Reviewer 2:
- Section 2.3. It is essential to quantify the concentration of Cefriaxone in gold nanoparticles by HPLC.
Response: As per the suggestion of the honorable reviewer, HPLC based quantification has been added in the revised MS.
- NMR analysis is required to evidence the capping and conjugation of Cefriaxone in gold nanoparticles?
Response: With all due respect, we would like to state that NMR is not available in our institute and has to be outsourced. Due to time constrain and current situation, we cannot do NMR at this point of time. However, we will keep this suggestion of honorable reviewer for our next work.
- SEM analysis results is poor, it need to redo with high magnification.
Response: As per the suggestion of honorable reviewer, we have duly added the SEM analysis with higher magnification of 100000X in the revised MS.
- In case SEM presented in different magnification but why TEM showed only one picture?
Response: As per the suggestion of honorable reviewer, TEM images at different magnifications have been duly added in the revised MS.
- Why the author not presented EDS results, AED, XRD with size measuring calculation.
Response: We appreciate the suggestion of the honorable reviewer; we have measured the size by DLS and TEM that has been considered acceptable in our earlier researches [1-3]. However, EDS results, AED, XRD facility are not available at our institute and we have to outsource them. We humbly request the honorable reviewer to kindly accept our size calculation, as it is very difficult for us to add these analyses at this point of time.
- Khan, S.; Haseeb, M.; Baig, M.H.; Bagga, P.S.; Siddiqui, H.H.; Kamal, M.A.; Khan, M.S. Improved efficiency and stability of secnidazole—An ideal delivery system. Saudi J Biol Sci 2015, 22, 42–49.
- Khan, S.; Rizvi, S.M.D.; Avaish, M.; Arshad, M.; Bagga, P.; Khan, M.S. A novel process for size controlled biosynthesis of gold nanoparticles using bromelain. Mater Lett 2011, 159, 373–376.
- Shaikh, S.; Rizvi, S.M.D.; Shakil, S.; Hussain, T.; Alshammari, T.M.; Ahmad, W.; Tabrez, S.; Al-Qahtani, M.H.; Abuzenadah, A.M. Synthesis and Characterization of Cefotaxime Conjugated Gold Nanoparticles and Their Use to Target Drug-Resistant CTX-M-Producing Bacterial Pathogens. J Cell Biochem 2017, 118(9), 2802-2808.
- FTIR analysis not clearly stated the formation of the NPS.
Response: With all due respect to the honorable reviewer, we would like to state that nanoparticles formation is already confirmed in our study by using UV-Vis spectroscopy, TEM, and DLS. Here, FTIR is only provided to confirm the presence of ceftriaxone on gold nanoparticle surface by amide bonds peaks.

Round 2
Reviewer 1 Report
After exhaustive revision of the new version of the manuscript, I sincerelly consider that requires additional revision. Concretely, the major limitation is the poor quality of the Figures 3 (the boxes Record2 and Record3 should be deleted), 4 (not treated), 5 (a lack of homogeneity between the scale bars) and S1 (blurred images).
Author Response
First of all, we would like to thank honorable reviewer to provide valuable comments to further improve our article. We have made all the changes in the revised MS as per your valuable suggestions.
The major limitation is the poor quality of the Figures:
- Figure 3 (the boxes Record2 and Record3 should be deleted)
Reply: Boxes Record 2 and 3 have been deleted in the revised MS.
- Figure 4 (not treated)
Reply: Untreated original figures have been added in the revised MS.
- Figure 5 (a lack of homogeneity between the scale bars)
Reply: Homogeneity of scale bars have been maintained in the revised figure 5.
- Figure S1 (blurred images).
Reply: Figure S1 has been replaced by a new clear image.

Reviewer 2 Report
accept
Author Response
First of all, we would like to thank honorable reviewer to accept our responses to the comments provided in round 1. We have duly checked the article for any grammatical, typographical and spelling mistakes in the revised MS.
